# Clinical Features and Management of Umbilical Endometriosis: A 30 Years’ Monocentric Retrospective Study

**DOI:** 10.3390/ijerph192416754

**Published:** 2022-12-14

**Authors:** Dhouha Dridi, Laura Buggio, Agnese Donati, Francesca Giola, Caterina Lazzari, Massimiliano Brambilla, Francesca Chiaffarino, Giussy Barbara

**Affiliations:** 1Gynecology Unit, Fondazione IRCCS Ca’ Granda Ospedale Maggiore Policlinico, 20122 Milan, Italy; 2Department of Clinical Sciences and Community Health, University of Milan, 20122 Milan, Italy; 3Plastic Surgery Service, Gynecology Unit, Fondazione IRCCS Ca’ Granda Ospedale Maggiore Policlinico, 20122 Milan, Italy

**Keywords:** endometriosis, umbilical endometriosis, pain, treatment, surgery

## Abstract

Introduction: Umbilical endometriosis (UE) is defined as the presence of endometrial-like tissue within the umbilicus and represents around 0.5–1% of all cases of endometriosis. UE is classified into primary or secondary UE. In this retrospective study, we aimed to assess symptoms, signs, recurrence rate of treated lesions, psychological wellbeing and health-related quality of life in women with UE. Material and methods: We retrospectively reviewed all cases of women diagnosed with UE in the period 1990–2021 in our center. Post-operative recurrence of UE was considered as the reappearance of the umbilical endometriotic lesion, or as the recurrence of local symptoms in the absence of a well-defined anatomical recurrence of the umbilical lesion. Moreover, participants were invited to fill in standardized questionnaires on their health conditions. Results: A total of 55 women with histologically proven UE were assessed in our center during the study period. At time of diagnosis, local catamenial pain and swelling were reported by 51% and 53.2% of women, respectively. A total of 46.8% of women reported catamenial umbilical bleeding. Concomitant non-umbilical endometriosis was identified in 66% of cases. As regards the treatment of UE, 83.6% of women underwent an *en-bloc* excision with histological confirmation of UE. During the follow-up period, 37 women (67.3%) agreed to undergo a re-evaluation. Recurrence of either umbilical symptoms, or umbilical nodule, was observed in 27% of patients, 11% of which did not receive post-operative hormonal therapy. Specifically, a recurrence of the umbilical endometriotic lesion was observed only in two women. Among the 37 women which we were able to contact for follow-up, 83.8% were satisfied with the treatment they had received. Conclusions: The high rate of patient satisfaction confirmed that surgical excision should be considered the gold standard treatment for umbilical endometriosis. Future studies should investigate the role of post-operative hormonal therapy, particularly in reducing the risk of symptom recurrence.

## 1. Introduction

Among extrapelvic endometriosis localizations, which occur in a distant site from the reproductive organs, abdominal wall endometriosis (AWE) is reported to be the most frequent [1].

Umbilical endometriosis (UE), or Villar’s nodule, first described by Villar in 1886, is defined by the presence of endometrial-like tissue within the umbilicus. This form represents around 0.5–1% of all cases of endometriosis [2,3,4], about 0.4–4% of extragenital cases of endometriosis [5,6] and about 21% of cases of AWE [7]. Clinically, UE presents with a red, purple, or black umbilical nodule, with a diameter ranging from 0.5 to 3.5 cm, causing pain, swelling and bleeding in the umbilical area (Figure 1) [8,9]. All these symptoms are exacerbated during the menstrual period [10,11,12].

Umbilical endometriosis is classified into primary or secondary UE, depending on the surgical history of the affected women. Primary UE, representing about 70% of all cases, arises without a surgical history, whereas secondary UE occurs on scar tissue following abdominal procedures at laparoscopy or laparotomy [6].

The distinction between primary and secondary UE appears important from a pathogenic perspective [13,14]. Over time, the following theories for UE have been developed: (a) migration of refluxed endometrial cells through the abdominal cavity; (b) dissemination of endometrial cells through the lymphatic system; (c) embryonic remnants in the umbilical fold (e.g., the urachus and umbilical vessels) [8,15,16].

In a recent systematic review, our research group suggested the following modified pathogenic theory for the origin of UE: primary umbilical endometriosis may originate from the implantation of regurgitated endometrial cells conveyed by the clockwise peritoneal circulation up to the right hemidiaphragm and funneled towards the umbilicus by the falciform and round liver ligaments [7].

Clinical recommendations for the management of extragenital endometriosis have been recently proposed by Hirata et al. [17]. The authors suggest radical surgery with wide local excision as the primary treatment for UE, although its long-term efficacy remains undefined. Medical therapy is supported by limited data and may not be curative. No study has directly compared medical and surgical treatment (Hirata et al.) [17].

Based on this background, we aimed to provide a broad perspective on umbilical endometriosis, performing a retrospective study to assess symptoms, signs, recurrence rate of treated lesions, psychological wellbeing and health-related quality of life in women with UE.

Our main objective was to investigate women’s satisfaction after UE treatment during a long-term follow-up. The secondary study objective was the evaluation of the effectiveness of different treatment modalities (i.e., surgical versus pharmacological versus combined therapy), particularly in terms of symptoms and lesion recurrence.

## 2. Materials and Methods

We retrospectively reviewed all cases of women diagnosed with UE between January 1990, and April 2021, in our tertiary-level referral center for the treatment of endometriosis, Gynaecology Unit, Fondazione IRCCS Ca’ Granda, Ospedale Maggiore Policlinico, Milan, Italy. The study was approved by the local Institutional Review Board (Comitato Etico Milano Area 2, reference 617_2021bis). In addition, all women involved in the study provided signed informed consent to authorize the use of clinical data for research.

Post-operative recurrence of UE was considered as the reappearance of the umbilical endometriotic lesion, or as the recurrence of local symptoms, such as swelling, pain, and bleeding, without a well-defined anatomical recurrence of the umbilical lesion.

The 55 women with UE were contacted by phone by two investigators (D.D. and F.G.) and invited to participate in the study and fill in standardized questionnaires on their health condition. Moreover, they were asked about pain symptoms, recurrence of lesions, and degree of satisfaction with the treatment received for UE, according to a five-category scale (very satisfied, satisfied, neither satisfied nor dissatisfied, dissatisfied, very dissatisfied).

The following variables were retrieved from medical records: age at diagnosis, body mass index (BMI), medical, surgical, and obstetrical history, presence of concomitant endometriotic lesions other than UE, endometriosis-associated symptoms, diagnostic imaging, surgical treatment, hormonal therapy, length of follow-up, post-operative recurrence (dichotomous scale, “yes” = 1, “no” = 0), and time to recurrence (months). The Italian version of the Hospital Anxiety and Depression Scale (HADS) [18] was used to assess the women’s psychological dimensions. The questionnaire is divided into two seven-item scales, one for anxiety and one for depression. Each scale can be treated independently with scores ranging from 0 to 21, although a total score can be calculated. Higher scores indicated poorer psychological conditions.

Quality of life (QoL) was assessed with the Short Form-12 questionnaire (SF-12) [19,20]. This 12-item questionnaire investigated the women’s perceptions of general health status (i.e., physical functioning, role limitation due to health conditions, general health, vitality and mental health). Higher scores indicated a better quality of life.

Patients’ overall health condition was assessed with the Patients’ Global Impression of Change (PGIC) scale, a seven-level scale (really improved, much improved, little improved, no change, little worse, much worse, really worse). Pain symptoms were assessed with the Patients’ Global Impression of Severity (PGIS) scale, which is composed of five levels (none, little, a bit, much, very much) [21].

A frequency analysis was performed for the variables examined in the study. Continuous variables were assessed with mean and standard deviation (SD). Categorical variables were presented as absolute values and percentages and analyzed with the chi-squared or Fisher’s exact test. The Kaplan–Meier method was adopted to analyze time to post-surgical recurrence. The event data used were the date of surgical excision or start of medical treatment, the date of diagnosis of UE recurrence, or that of the last follow-up visit.

## 3. Results

A total of 55 women with histologically proven UE were assessed in our center during the study period. The demographic characteristics and clinical features of women enrolled in the study are reported in Table 1.

The mean age at diagnosis was 33.6 ± 7.4 years. A total of 34/54 women were nulliparae (63%; 95% CI 48.7–75.7). In addition, 9/54 women (16.6%; 95% CI 7.9–29.3) had previously had a vaginal delivery, while 11/54 (20.4%; 95% CI 10.6–33.5) had previously undergone a cesarean section.

Twenty-three/48 women (48%; 95% CI 33.3–62.8) had no history of abdominal surgery and were considered as presenting with a primary UE form, whereas 26 (54.2%; 95% CI 39.2–68.6) had previously undergone a laparotomy for reasons other than a cesarean section. In addition, three women had undergone umbilical hernia repair.

All women reported specific local symptoms as well as the presence of an umbilical nodule.

Local catamenial pain and swelling were reported in 24/47 women (51%; 95% CI 36.1–65.9) and 25/47 women (53.2%; 95% CI 38.1–67.9), respectively. A total of 46.8% of women complained of catamenial umbilical bleeding (22/47; 95% CI 32.1–61.9).

Concomitant non-umbilical endometriosis was identified in 66% of cases (bilateral ovarian endometriomas *n* = 7; left ovarian endometriomas *n* = 12; right ovarian endometriomas *n* = 7; peritoneal endometriosis *n* = 9; deep endometriosis *n* = 14; diaphragmatic endometriosis *n* = 6; hepatic endometriosis *n* = 1. More than one localization could be present at the same time).

A total of 41 women underwent surgical excision with histological confirmation of UE (83.6%; 95% CI 70.3–92.7). Among these, seven women had already undergone excision of the UE nodule. In addition, seven women were treated with hormonal therapy only (see Table 2).

Data regarding skin and subcutaneous ultrasonography of the umbilical region was available for 35 women. Computed tomography (CT) and magnetic resonance imaging (MRI) were performed, respectively, in two cases.

Among the 55 women with UE, 18 could not be contacted. The remaining 37 women (67.3%) agreed to undergo a re-evaluation in follow-up.

The follow-up period ranged from 11 months to 20 years. During this period, ten cases of recurrence (10/37, 27%¸ 95% CI 13.8–44.1) of either umbilical symptoms or umbilical nodule were observed. Specifically, three women reported local pain only, one reported both swelling and bleeding, three reported both pain and swelling, and three reported all three umbilical symptoms. Only two women reported the recurrence of the umbilical endometriotic lesion (2/37, 5.4%, 95% CI 0.6–12.2). According to the Kaplan–Meier analysis (Figure 1), the cumulative incidence of postoperative recurrence was 0.03 (0.002–0.13) at 6 months, 0.11 (0.04–0.24) at 12 months and 0.36 (0.17–0.55) at 240 months following surgery (see Figure 2). Among the ten women who presented with a recurrence of UE, three had not received post-operative hormonal therapy (11%).

Among the 37 women whom we were able to contact for follow-up, 31 were satisfied with the received treatment (*n* = 21 very satisfied, and *n* = 10 satisfied) (83.8%, 95% CI% 0.68–0.94), three women were neither satisfied nor dissatisfied, two women were dissatisfied, and one woman was very dissatisfied (Table 3).

Means and standard deviations for mental health variables (SF-12 score and HADS score) and number and percentages related to the assessment of women’s global conditions and symptoms (PGIC and PGIS) are shown in Table 3. Regarding the PGIC, a significant improvement in the overall health condition (“much improved” or “really improved”) was observed in 81% of women (30/37, 95% CI 64.8–92). Moreover, the PGIS showed that 51% of patients (19/37, 95% CI 34.4–68.1) had no symptoms, while 35.1% (13/37, 95% CI 20.2–52.5) reported few symptoms (see Table 3).

In our case series, 41/49 women underwent surgical treatment for UE. Only 7/49 patients received pharmacological treatment. This small number of patients did not allow us to make a direct comparison between surgical and pharmacological treatments. The clinical features of women with UE treated with pharmacological therapy and their degree of satisfaction are reported in Table 4.

## 4. Discussion

Our findings confirmed that about half (48%) of UE lesions were primary. This result supported the hypothesis that this particular form of endometriosis might originate from the implantation of refluxed endometrial cells conveyed by the clockwise peritoneal circulation up to the right hemidiaphragm and funneled toward the umbilicus by the falciform and round liver ligaments [7]. Thus, the pathogenesis of UE may differ from that of other AWE types, which are typically strongly associated with surgical scars [5,22]. Moreover, six women (11.1%) had concomitant umbilical and diaphragmatic or pleural endometriosis. This data further supported the above suggested pathogenic theory for UE [7]. However, despite being considerable, our percentage of primary UE was lower than that reported in a recent systematic review (i.e., about 70%) [7].

A recent systematic review on umbilical endometriosis [7] reported the results of 11 studies (10 retrospective and one prospective) and 14 case series. Overall, 232 UE cases were reported, with the number per study ranging from 1 to 96. Following Hirata’s study (2020) [6], the present retrospective study is the largest research on clinical characteristics and management of UE. However, Hirata’s study was multicentric, whereas our research is monocentric.

As previously reported [6], most women with UE are symptomatic (97.9%). Similarly, we observed that women with UE frequently complain of local pain or other local disturbing symptoms. Catamenial local pain and swelling were the most common complaints, whereas umbilical bleeding was reported less frequently.

In our case series, the vast majority of women (84%) underwent surgical treatment for UE. Only a small percentage of patients (14%) received hormonal treatment. The small number of patients treated with hormonal therapy did not allow us to compare medical and surgical treatments directly. However, considering post-surgical outcomes in terms of women’s satisfaction, we can conclude that surgical treatment may be considered the gold standard for treating and managing UE. Data presented in a large multicenter Japanese national survey on umbilical endometriosis [6] were consistent with our results, showing that the vast majority of women (90.6%) underwent successful surgical treatment with radical excision of the umbilical lesion.

Although data reported in the literature are limited, pharmacological therapy (combined oral contraception, progestins, GnRH agonists) should be a good option in women who refuse surgery or before surgery to ameliorate painful symptoms.

During a long follow-up period (up to 20 years), we observed postoperative recurrence of clinical symptoms and/or umbilical lesions in ten cases (27%). Only 3 women out of ten had been surgically treated for UE in a referral center by a surgeon using the *en-bloc excision* technique. This could explain the high rate (27%) of clinical recurrence. Specifically, in all these cases the severity of painful symptoms (Numerical Rating Scale, NRS) was low (NRS mean and SD: 4.75 ± 2.12).

However, in our study only two women reported recurrence of the umbilical endometriotic lesion (2/37, 5.4%, 95% CI 0.6–12.2). This result was similar to that reported by Hirata et al. [6], who reported a low rate of post-operative umbilical lesion recurrence.

Pre-surgical soft tissue ultrasound is a valuable procedure to improve surgical resection success [23,24,25]. Numerous authors previously reported that *en-bloc excision* of UE lesions, including the peritoneum, and the subsequent reconstruction of the umbilicus, guarantees the radicality of the procedure [6,7,17]. According to our data and previous observations [7], post-operative UE lesion recurrence seems to be substantially lower for UE than for ovarian endometriosis. For this reason, in cases of isolated UE (with non-concomitant pelvic endometriosis or other extragenital forms of endometriosis), the role of post-operative hormonal treatment in reducing the risk of lesion recurrence is still undefined and should be evaluated in future studies.

Despite the high rate of clinical UE recurrence (27%), women reported a high satisfaction rate with treatment (83.8%). The low rate of UE lesion recurrence (5.4%), as well as the PGIC and PGIS scores, indicated a significant improvement in the overall health condition in about 80% of women and the absence, or the presence of, minimal symptoms in about 86% of patients, may explain the observed high rate of women’s satisfaction.

Ultrasonography may be considered the first-level diagnostic tool [23,24,25,26]. However, as ultrasound is highly operator-dependent and requires specific skills, and should be performed by trained and experienced practitioners. Therefore, MRI and CT should be reserved only for selected cases [27].

Our study reported no cases of malignant transformation of the umbilical nodule. However, according to Hirata et al. (2020), UE’s risk of malignant transformation is about 3% [6]. In the literature, only 4 cases of malignant transformation of UE are reported (one endometrioid adenocarcinoma, one clear cell adenocarcinoma and two adenocarcinomas) [28,29,30].

Our study had some limitations. Firstly, most of the data were collected retrospectively from existing medical records. This methodology carries the typical limitations of recorded patient information, including a high rate of unreported information and possible errors. Moreover, as umbilical endometriosis is a rare localization of the disease, data collection included cases treated in our center in the last 30 years. Such a large time frame made it difficult to contact many women.

## 5. Conclusions

Considering the low risk of UE lesion recurrence in cases of *en-bloc* surgical removal, the high rate of patient satisfaction, as well as the reported 3% risk of malignant transformation [6,7,17], our results confirmed that surgical excision should be considered the gold standard for the treatment of umbilical endometriosis. Future studies should investigate the role of post-operative hormonal therapy, particularly in reducing the risk of symptom recurrence.

## Figures and Tables

**Figure 1 ijerph-19-16754-f001:**
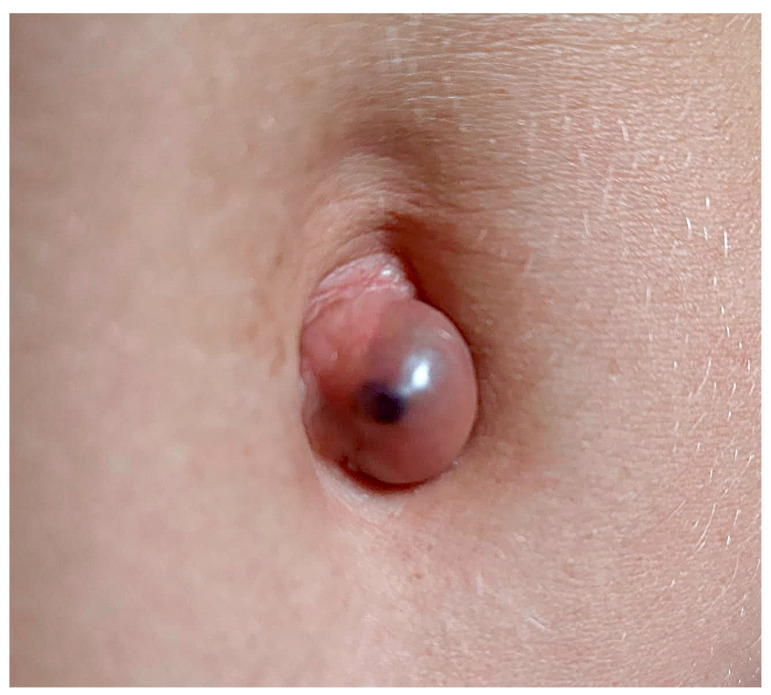
Umbilical endometriosis nodule.

**Figure 2 ijerph-19-16754-f002:**
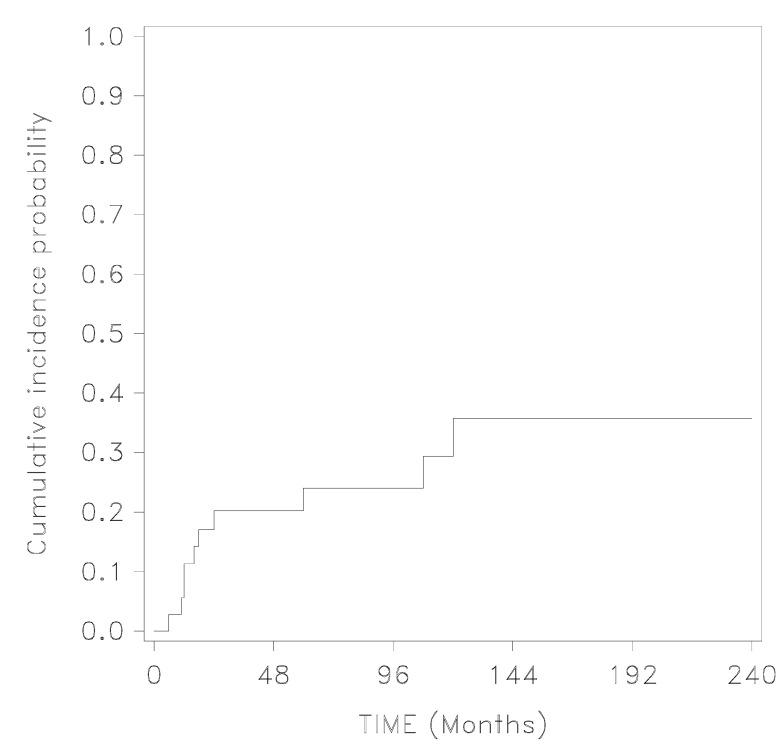
Cumulative incidence probability of recurrence rate after surgical treatment of umbilical endometriosis.

**Table 1 ijerph-19-16754-t001:** Demographic and clinical characteristics of women enrolled in the study.

Characteristics	Enrolled Women % (*n*/Total ^a^)
**Age (years)**	45 ± 9.6
**Smoke**	
No	74 (34/46)
Yes	26 (12/46)
**BMI (Kg/m^2^)**	22.7 ± 3.4
**Age at diagnosis (years)**	33.6 ± 7.4
**Obstetric history**	
Nulliparous	63 (34/54)
Multiparous	37 (20/54)
Vaginal delivery	17 (9/54)
Caesarean section	20 (11/54)
**Previous abdominal surgery **	
None	33.4 (16/48)
Abdominal surgery	65 (32/48)
LPS *	8 (4/48)
LPT **	54 (26/48)
Both LPS and LPT	2 (1/48)
Only onfaloplasty for umbilical hernia	2 (1/48)
**Medical therapy prior to UE surgical treatment**	
None	33 (17/52)
Estroprogestins	62 (32/52)
Progestins	6 (3/52)
Others	0
**Umbilical pain**	
None	13 (6/47)
Non catamenial	15 (7/47)
Catamenial	51 (24/47)
Both	21 (10/47)
**Umbilical bleeding**	
None	36 (17/47)
Non catamenial	17 (8/47)
Catamenial	47 (22/47)
Both	0
**Umbilical swelling**	
None	62 (17/47)
Non catamenial	6 (3/47)
Catamenial	53 (25/47)
Both	4 (2/47)

^a^ The total of women was not always 55, due to the presence of missing data. * One woman had both laparoscopy and onfaloplasty. LPS = laparoscopy. ** Two women had both laparotomy and onfaloplasty. LPT = laparotomy. Data are reported as mean ± SD, or percentage (*n*/total). BMI = Body Mass Index.

**Table 2 ijerph-19-16754-t002:** Treatment modality and clinical characteristics of UE.

	Enrolled Women % (*n*/Total ^a^)
**Surgical treatment**	84 (41/49)
**Medical treatment**	14 (7/49)
**None**	2 (1/49)
**Postoperative medical therapy**	
No	23 (9/40)
Yes	78 (31/40)
**Concomitant laparoscopy**	
No	64 (25/39)
Yes	36 (14/39)
**Presence of endometriosis other than UE**	
No	4 (2/53)
Yes	66 (35/53)
No, but not performed laparoscopy	30 (16/53)
**^b^ UE recurrence after surgery**	
No	73 (27/37)
Lesion and/or symptoms recurrence	27 (10/37)
Only lesion recurrence	5.4 (2/37)

^a^ The total number of women was not always 55, due to the presence of missing data. ^b^ Only 37 women underwent follow-up. Data are reported as percentage (*n*/total).

**Table 3 ijerph-19-16754-t003:** Follow up of 37 women with umbilical endometriosis. Degree of satisfaction with treatment and results of the standardized questionnaires on the women’s health conditions.

	Enrolled Women ^a^ % (*n*)
**Degree of satisfaction with treatment**	
Very satisfied	56 (21)
Satisfied	27 (10)
Neither satisfied nor dissatisfied	8 (3)
Dissatisfied	5 (2)
Very dissatisfied	3 (1)
**Global condition since beginning of therapy (PGIC)**	
Really improved	29 (11)
Much improved	51 (19)
Little improved	11 (4)
Same	3 (1)
Little worse	0
Much worse	3 (1)
Really worst	3 (1)
**Symptoms in the last 4 weeks (PGIS)**	
None	51 (19)
Little	35 (13)
A bit	8 (3)
Much	3 (1)
Very much	3 (1)
**HADS**	
Anxiety Score	7.4 ± 3.8
Depression Score	5.9 ± 4.1
Total Score	6.6 ± 4.0
**SF-12**	
Physical Component Summary Score	45.18 ± 9.42
Mental Component Summary Score	44.81 ± 10.17

^a^ The total of women is not always 37, due to the presence of missing data. Data is reported as mean ± SD, or percentage and number. PGIC = Patients’ Global Impression of Change. PGIS = Patients’ Global Impression of Severity. HADS = Hospital Anxiety and Depression Scale. SF-12 = Short form-12.

**Table 4 ijerph-19-16754-t004:** Clinical features of women with UE treated with pharmacological therapy and their degrees of satisfaction.

Patient ID Code	Age at Diagnosis	Number of Pregnancies	Symptoms	Previous Surgery	Concomitant Non-UE	Type of Pharmacological Treatment	Satisfaction with Treatment *	PGIC **	PGIS ***
1	NA	0	Cyclic umbilical bleeding, swelling and local pain	LPT hysterectomy (myomas)	Left ovarian endometriomas	Estroprogestins	Very satified	Much improved	None
2	34	0	Cyclic umbilical bleeding, swelling and local pain	Appendicectomy	None	Progestins	Satisfied	Much improved	Little
3	30	0	Cyclic umbilical swelling and local pain	None	Left ovarian endometrioma and deep endometrosis	Estroprogestins	Very satified	Much improved	None
4	NA	0	Cyclic umbilical bleeding, swelling and local pain	Onfaloplasty (umbilical hernia)	None	NA	Neither satisfied nor dissatisfied	No change	Little
5	52	1	Cyclic umbilical swelling and local pain	None	Thoracic endometriosis	GnRH agonist	Very satified	Really improved	None
6	43	1	Cyclic umbilical swelling	Onfaloplasty (umbilical hernia). LPS right ovarian endometrioma excision	Right ovarian endometrioma	Progestins	Satisfied	Much improved	A bit
7	35	2	Cyclic umbilical bleeding, swelling and local pain	2 Cesarian sections	None	NA	Neither satisfied nor dissatisfied	No change	None

NA: data not available. * According to a five-category scale (very satisfied, satisfied, neither satisfied nor dissatisfied, dissatisfied, very dissatisfied). ** Patients’ Global Impression of Change (PGIC) scale, a seven-level scale (really improved, much improved, little improved, no change, little worse, much worse, really worse). *** Patients’ Global Impression of Severity (PGIS) scale, which is composed of five levels (none, little, a bit, much, very much).

## Data Availability

All data generated or analyzed during this study are included in this published article.

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
