# Peer review of "Clinical Features and Management of Umbilical Endometriosis: A 30 Years’ Monocentric Retrospective Study"

_ijerph, 2022, doi:10.3390/ijerph192416754_

Round 1

Reviewer 1 Report

Major comments

1.      It is well established that surgical treatment is the common first line management for UE, though the recent treatment plan includes hormonal therapy too. The present manuscript lacks the novelty, authors may focus the comparison between surgical/ medication outcome and patient satisfaction that would help better understanding and author’s claim (line # 259- 263).

2.       The present manuscript is insufficient with review of the current literature and the author mainly focused only two papers (Reference # 6 &7) throughout the manuscript. There are many case study report available on UE and clinical characteristics eg., Theunissen and IJpma, 2015; Taniguchi et al., 2016; Makena et al., 2020 etc.,

3.       This manuscript needs major revision; the reference cited in the text are not in the same format see line # 61, Line # 64 & Line #214  The reference cited in the text (Line #61) is not same in the reference listed (Line # 299). Line #249- 250 reference missing, Table 1&2 heading are not aligned properly. Reference 19 is not cited in the text.

Author Response

We acknowledge that Reviewer 1 is right. One of our secondary objectives was the comparison between medical treatment and surgical treatment for umbilical endometriosis. However, the sample of women treated with medical therapy without surgery was too small, as compared to the sample of women treated surgically. This does not allow us to perform a comparison between the two treatments.  See page 8, lines 208-212. We have added to the Manuscript a new table (Table 4), showing the clinical features of women with UE treated with pharmacological treatment.

Reviewer 2 Report

Great work done with a well-written manuscript. 

Could you provide the questionnaire as supplementary materials please? Wondering if any detailed description of different levels of satisfaction. 

Author Response

We thank the reviewers for the positive comment. We provided the questionnaire used for the study as supplementary material.

Reviewer 3 Report

We congratulate the authors about their article on Umbilical Endometriosis. We suggest the following edits:

Line 40: replace "as" by "by"

Line 55 to 58: how the authors can generalize their theories on other extrapelvic endometriosis locations? do they think that every location of endometriosis has a different pathogenic process?

Line 70 The authors state that their secondary objective is about the efficacy of different treatments. The conclusion states that surgical correction is superior and discuss patient's satisfaction, however tables should be added to delineate this result. In table 3, it is mentioned very briefly.

Line 119: Table 1 is not clear. Tabulation should be adjusted in the title %(n/total) should be adjusted.

Line 146: same as table 2.

Author Response

Line 40: replace "as" by "by".

AR: Done (see page1, line 40).

Line 55 to 58: how the authors can generalize their theories on other extrapelvic endometriosis locations? do they think that every location of endometriosis has a different pathogenic process?

AR: In a recent paper, we suggested a modified pathogenic theory for the origin of UE: primary umbilical endometriosis may originate from the implantation of regurgitated endometrial cells conveyed by the clockwise peritoneal circulation up to the right hemidiaphragm and funneled towards the umbilicus by the falciform and round liver ligaments (see Introduction section, page 2, lines 57-61).

As discussed at page 8, lines 208-216 our findings confirm that about half (48%) of UE lesions were primary and this result may support the hypothesis that this particular form of endometriosis may originate from the implantation of refluxed endometrial cells conveyed by the clockwise peritoneal circulation up to the right hemidiaphragm and funneled toward the umbilicus by the falciform and round liver ligaments. Thus, the pathogenesis of UE may differ from that of other AWE types, typically strongly associated with surgical scars.

Line 70 The authors state that their secondary objective is about the efficacy of different treatments. The conclusion states that surgical correction is superior and discuss patient's satisfaction, however tables should be added to delineate this result. In table 3, it is mentioned very briefly.

AR: We agree with the Reviewer. As reported at page 8, lines 208-212, 41/49 women underwent surgical treatment for UE. Only 7/49 patients received pharmacological treatment. This small number of patients did not allow us to make a direct comparison between surgical and pharmacological treatment. Clinical features of women with UE treated with pharmacological therapy and their degree of satisfaction are reported in a new table (Table 4).

Line 119: Table 1 is not clear. Tabulation should be adjusted in the title %(n/total) should be adjusted.

AR: Done

Line 146: same as table 2.

AR: Done

Round 2

Reviewer 1 Report

The revised manuscript addressed all the question raised, I recommend this manuscript for publication.